# C5T5: Controllable Generation of Organic Molecules with Transformers

## Abstract

Methods for designing organic materials with desired properties have high potential impact across fields such as medicine, renewable energy, petrochemical engineering, and agriculture. However, using generative models for this task is difficult because candidate compounds must satisfy many constraints, including synthetic accessibility, intellectual property attributes, "chemical beauty" (Bickerton et al., 2012), and other considerations that are intuitive to domain experts but can be challenging to quantify. We propose C5T5, a novel self-supervised pretraining method that works in tandem with domain experts by making zero-shot select-and-replace edits, altering organic substances towards desired property values. C5T5 operates on IUPAC names—a standardized molecular representation that intuitively encodes rich structural information for organic chemists but that has been largely ignored by the ML community. Our technique requires no edited molecule pairs to train and only a rough estimate of molecular properties, and it has the potential to model long-range dependencies and symmetric molecular structures more easily than graph-based methods. We demonstrate C5T5's effectiveness on four physical properties relevant for drug discovery, showing that it learns successful and chemically intuitive strategies for altering molecules towards desired property values.

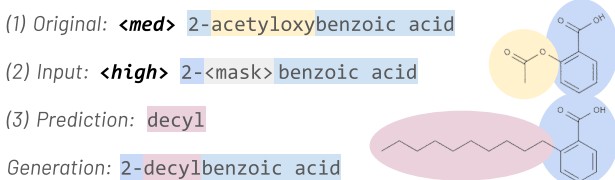

Figure 1: **Increasing a molecule's octanol-water partition coefficient with C5T5**. (1) A molecular fragment (`acetyloxy`) is identified in a molecule of interest. (2) The molecular fragment is replaced with a mask token (`<mask>`) and the property value is set to the desired bucket (`<high>`). (3) Sampling from C5T5 produces a new fragment (`decyl`). Substituting in the fragment yields a new molecule. The long chain of carbons added to the molecule increases its solubility in octanol while decreasing its solubility in water.

## 1 Introduction

Organic molecules are used in countless applications across human society: as medicines, industrial chemicals, fuels, pesticides, plastics, television screens, solar cells, and many others. Traditionally, new molecules are designed for particular tasks by hand, but the space of all possible molecules is so vast (e.g. the total number of drug-like molecules may be as high as $10^{60}$) that most useful materials are probably still undiscovered (Reymond et al., 2012). To automate materials discovery, domain experts have turned to high-throughput screening, in which a large library of potentially useful molecules is generated heuristically, and the most promising molecules are chosen for further study using computational models that estimate how effective each substance will be for the target application (Hughes et al., 2011). Unfortunately, even high-throughput methods can still only screen a tiny fraction of all possible molecules.

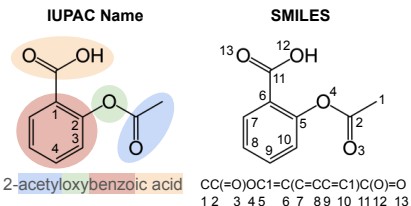

Figure 2: **Visual representations of IUPAC names and SMILES.** Tokens in IUPAC names correspond to well-known functional groups and moieties. In contrast, tokens in SMILES correspond to individual atoms and bonds.

Generating molecules directly with machine learning addresses this limitation, but *de novo* generation can be of limited use in domains like drug discovery, where experts' intuitions about structure-activity relationships and external factors like patentability are important to consider in the design process. These constraints can often be expressed by providing known portions of the molecular structure; for example a domain expert may be interested in a particular scaffold because it has favorable intellectual property attributes, or certain parts of a drug may be needed for the desired biological activity, while other parts can be modified to increase bioavailability.

To address this real-world setting, we consider the problem of learning to make localized modifications to a molecule that change its physical properties in a desired way. We propose C5T5: Controllable Characteristic-Conditioned Chemical Changer with T5 (Raffel et al., 2019), a novel method for generative modeling of organic molecules that gives domain experts fine-grained control over the molecular optimization process while also providing more understandable predictions than prior methods (Figure 1). Our two key contributions are 1) recasting molecular modeling as language modeling on the semantically rich **IUPAC name** base representation, and 2) the development of a novel **conditional language modeling** strategy using transformers that supports targeted modifications to existing molecules.

**IUPAC Names.** The IUPAC naming system is a systematic way of naming organic molecules based on functional groups and moieties, or commonly occurring clusters of connected atoms that have known chemical behaviors. Organic chemists have discovered countless chemical reactions that operate on functional groups, and they use these reactions to develop synthesis routes for novel molecules. Despite this, existing generative methods for organic molecules have ignored IUPAC names as a representation, instead opting for atom-based representations like SMILES (Weininger, 1988) and molecular graphs (Duvenaud et al., 2015). See Figure 2 for a comparison of these representations. We argue for several advantages of IUPAC names in Section 3.1. To the best of our knowledge we are the first to use IUPAC names as a base representation for molecular modeling.

**Self-Supervised Objective for Zero-Shot Editing.** To enable targeted modifications of molecules without predefined edit pairs, we train transformers with a conditional variant of a self-supervised infilling task, where the model must replace masked-out tokens in the IUPAC name. As described in Section 3.2, we condition the model by prepending IUPAC names with discretized molecular property values; the model then learns the conditional relationships between the property value and molecular structure. To the best of our knowledge, C5T5 is the first method to use conditional infilling for select-and-replace editing; we anticipate this method could be broadly applied in other controlled generation contexts, such as modeling affect, politeness, or topic of natural language (Ghosh et al., 2017; Ficler & Goldberg, 2017; Niu & Bansal, 2018; Keskar et al., 2019).

As we show in Section 4, C5T5 is able to make interpretable targeted modifications to molecules that lead to desired changes across several physical properties important in drug design.

## 2 RELATED WORK

**Modeling.** A number of machine learning methods have been developed for the task of designing organic molecules, but most do not allow a user to make targeted modifications to a molecule. Some methods, like generative adversarial networks and unconditional sequence models, provide no

control over a generated molecule's structure (Grisoni et al., 2020; Guimaraes et al., 2017; Sanchez-Lengeling et al.; De Cao & Kipf, 2018; Kajino, 2019), and are therefore more useful for generating candidate libraries than optimizing a particular molecule. Other methods, like variational autoencoders or sequence models that are conditioned on a base molecule, allow specifying that generated molecules should be similar to a starting molecule in some learned space, but there is no way to specifically target a certain part of the molecule to modify (He et al., 2021b; Jin et al., 2019; Shin et al., 2021; Yang et al., 2020; Kotsias et al., 2020; Gómez-Bombarelli et al., 2018; Lim et al., 2018; Dollar et al., 2021; Liu et al., 2018; Jin et al., 2018; Maziarka et al., 2020; Olivecrona et al., 2017; Bagal et al., 2021; You et al., 2018; Shi* et al., 2020). Recognizing the importance of leveraging domain experts' intuition about structure-activity relationships, several methods, published mostly in chemistry venues, have explored constraining generated molecules to contain a scaffold, or a subgraph of the full molecular graph (Li et al., 2019; Lim et al., 2020; Maziarz et al., 2021). However, these methods append to scaffolds arbitrarily instead of allowing domain experts to specify which part of the molecule they would like to modify or append to, limiting their utility for human-in-the-loop molecular optimization.

A few methods have explored allowing targeted modifications, where a domain expert can mask out a portion of a starting molecule and ask the model to replace the mask with a novel side chain (Arús-Pous et al., 2020; Langevin et al., 2020; He et al., 2021a). These methods are limited because they only support masking parts of the molecule that can be truncated by cutting a single bond, and because they require a dataset of paired molecules (scaffolds & decorators) that must be constructed using hand-crafted rules. In contrast, C5T5 learns in an entirely unsupervised fashion and therefore requires no paired data; the only limit to what can be masked is what can be represented using IUPAC tokens.

**Representation.**   Existing methods all use SMILES (or a derivative representation) or graphs to represent molecules. There are a number of drawbacks to using the SMILES representation: a small change in a molecule can lead to a large change in the SMILES string (Jin et al., 2018); flattening the graph into a list of atoms artificially creates variable- and long-range dependencies between bonded atoms; and it is difficult to reason about common substructures, because the same structure can be represented in many different ways depending on how the graph was flattened. And although graphs seem like a natural representation for molecules, graphs do a poor job encoding symmetry, long-range interactions between atoms that are many bonds apart but nearby in 3D space, and long-range interactions that arise from conjugated systems (Duvenaud et al., 2015). C5T5 operates instead of IUPAC names, which we argue in Section 3.1 is a more suitable representation for molecular optimization because tokens have much more semantic meaning. See Appendix A for more details on how C5T5 relates to prior work.

**Transformers for Molecular Modeling**   Outside of molecular optimization, transformers have found a number of applications in molecular modeling tasks, including property prediction (Wang et al., 2019; Rong et al., 2020), chemical reaction prediction (Schwaller et al., 2019), retrosynthesis (Karpov et al., 2019) and generating proteins (Elnaggar et al., 2020; Grechishnikova, 2021). A few works have explored using transformers for generative modeling of organic molecules (He et al., 2021b; Shin et al., 2021; Dollar et al., 2021). Some works have also proposed using transformers for scaffold-conditioned generative modeling (He et al., 2021a; Bagal et al., 2021). This work extends these efforts by proposing a simple yet effective training and zero-shot adaptation method, and by using IUPAC names instead of SMILES strings.

**IUPAC Names**   Although we are unaware of prior work using IUPAC names as a base representation for molecular modeling, several works have explored using machine learning to convert between IUPAC names and other molecular representations (Rajan et al., 2021; Handsel et al., 2021; Krasnov et al., 2021).

## 3   METHOD

Molecular optimization is a difficult problem because it requires modifying a molecule that already satisfies a number of requirements. Modifications need to improve a particular aspect of the molecule without degrading its performance on other metrics, and without making it too difficult

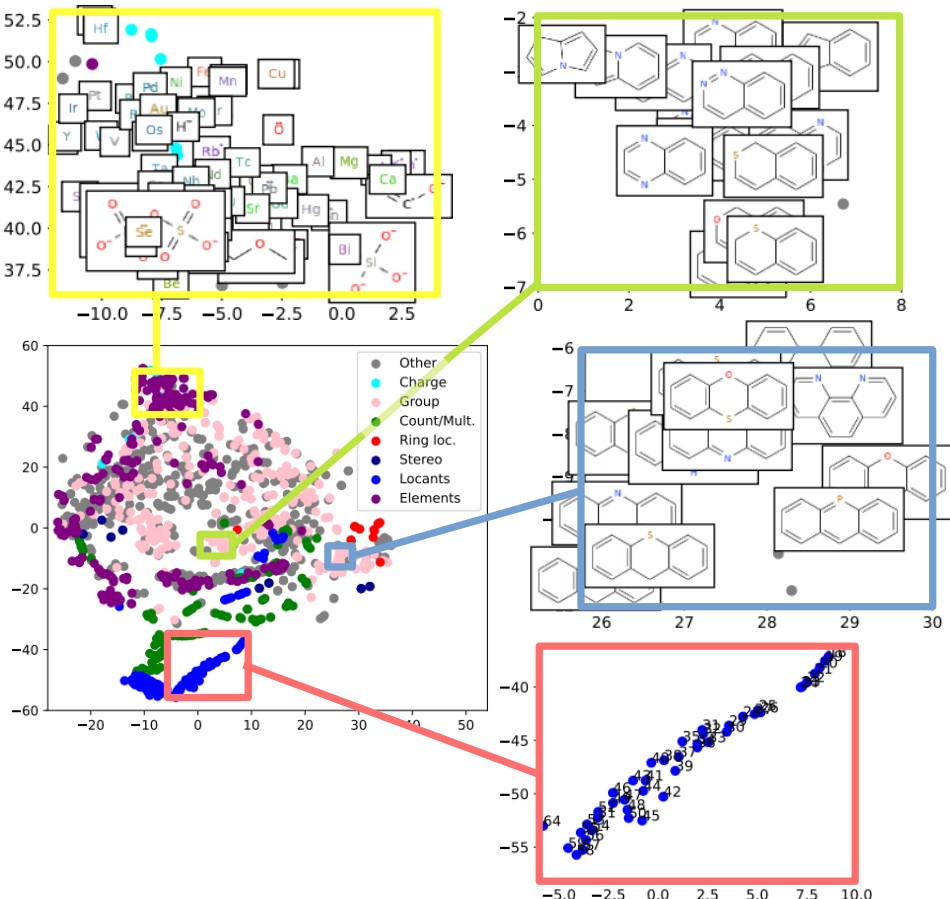

Figure 3: **T-SNE visualization of the word2vec embedding space.** "Charge": tokens that indicate formal charge. "Group": functional groups and moieties. "Count/Mult": multipliers. "Ring loc.": fused-ring locants. "Stereo": stereochemistry markers. "Locants": simple locants. "Elements": single-atom tokens. As shown, the 2D location of tokens carries high semantic meaning; for example, locants are not only collocated, but are approximately in order.

to synthesize. We argue that by using IUPAC names (Section 3.1) and by allowing users to target particular parts of a molecule to modify (Section 3.2), C5T5 has the potential to support human-in-the-loop molecular editing that complements domain experts' intuitions about structure-activity relationships and synthetic accessibility.

## 3.1 IUPAC NAMING

The International Union of Pure and Applied Chemistry (IUPAC) publishes a set of rules that allow systematic conversion between a chemical structure and a human-readable name (Favre & Powell, 2013). For example, 2-chloropentane refers unambiguously to five carbons ("pent") connected by single bonds ("ane") with a chlorine atom ("chloro") bonded to the second carbon from one end ("2-"). IUPAC names are used ubiquitously in scholarly articles, patents, and educational materials. In contrast to other linear molecular representations like SMILES and its derivatives, where single tokens mostly refer to individual atoms and bonds, tokens in IUPAC names generally have a rich semantic meaning. For example, the token "ic acid" denotes a carboxylic acid, which is a common functional group that has well-known physical and chemical properties; there are many known chemical reactions that either start with or produce carboxylic acids. Other tokens denote additional functional groups (e.g. "imide," "imine," "al," "one"), locants (e.g. "1," "2," "N"), which indicate connectivity, alkanes (e.g. "meth," "eth," "prop"), which denote the lengths of carbon chains, poly-cyclic rings (e.g. "naphthalene," "anthracene"), stereochemistry markers ("R," "S"), and multipliers

(e.g. "di," "tri"), which concisely represent duplicated and symmetric structures. Figure 2 shows the relationships between IUPAC names, graph representations, and SMILES.

For molecular optimization, C5T5 supports qualitatively different molecular edits compared to graph- and SMILES-based methods by virtue of its use of IUPAC names: editing a locant token corresponds to moving a functional group along a carbon backbone or changing the connectivity of a fused ring system; and editing a multiplier token corresponds to creating or eliminating duplicated and symmetric structures. For example, changing "ethylbenzene" to "hexaethylbenzene" replicates the ethyl structure around the entire benzene ring with a single token edit. These sorts of modifications require much more extensive editing for SMILES- and graph-based methods.[1]

We argue that IUPAC names are especially attractive for molecular optimization, since the process requires interaction between the algorithm and a domain expert, so interpretability is paramount. Compared to graph- or SMILES-based models, C5T5 makes predictions that can be traced back to moieties and functional groups that domain experts are more likely to understand, trust, and know how to synthesize than either arbitrary collections of atoms and bonds or motifs decided upon by machine learning practitioners.

In addition to improved interpretability, we argue that using IUPAC names has advantages purely from the standpoint of modeling data, since moving from SMILES to IUPAC names is akin to moving from a character-based to a word-based sequence model. Modeling at this higher level of abstraction enables the network to direct more of its capacity to structure at the relevant semantic level, instead of relearning lower-level details like the specific atomic composition of functional groups. In this vein, we demonstrate the potential of IUPAC names by learning word2vec representations of IUPAC name tokens (Mikolov et al., 2013), drawn from a list of over 100 million names in the PubChem repository (Kim et al., 2016) and tokenized using a list of tokens in OPSIN—an open-source IUPAC Name parser library (MIT License) (Lowe et al., 2011). For example, as shown in Figure 2, the chemical "2-acetyloxybenzoic acid" gets tokenized to ["2", "-", "acet", "yl", "oxy", "benzo", "ic acid"]. As with natural language modeling, we find that the embedding space learned by word2vec encodes the semantic meaning of the tokens, as shown in Figure 3. Different classes of tokens tend to be clustered together, and similar tokens within clusters are located nearby. For example, aromatic compounds with two rings are clearly separated from those with three, locants are ordered roughly correctly from 1 to 100, and multiplier tokens are also roughly in order (zoom not shown). Following Mikolov et al. (2013), we also find that simple arithmetic operations in the embedding vector space correspond to semantic analogies between tokens. For example, the nearest neighbor of "phosphonous acid" - "nitrous acid" + "nitroso" is the embedding for "phosphoroso."[2] The nearest neighbor of "diphosphate" - "disulfate" + "sulfate" is "phosphate." Likewise for "selenate" - "tellurate" + "tellurite" being closest to "selenite."

## 3.2 Conditional Language Modeling with Transformers

We now present the C5T5 objective, which trains a model to alter the properties of a molecule through localized edits. Importantly, this behavior does not require training on human-defined edit pairs, a strategy limited by either a fixed set of hand-specified chemical alterations or an expensive experimentally-derived training set. Instead, this editing behavior emerges as a zero-shot side-effect of our conditional language modeling objective, requiring only a simple forward pass using our pretrained model without additional gradient updates.

Given a property value $P$, a fragment of a molecule $F$, and the rest of the molecule $C$ (the context), we wish to learn the conditional distribution $P(F|C, P)$. Then, for a new molecule, one could alter the molecule towards a desired property value by redacting the original $F$, changing $P$ to $P'$ and sampling a new $F' \sim P(F|C, P')$. Intuitively, this asks our model what kinds of molecular fragments the model would expect given the context and the new property value.

---

[1]Moving the attachment point of a functional group requires only a small edit of a graph (i.e. changing one bond), but most graph-based molecular generation methods sequentially generate one node at a time, followed by any bonds that connect the new node to the molecular graph so far. Moving a side chain therefore requires removing the entire chain and regenerating it node by node.

[2]ignoring the embedding for "nitroso"

To learn this conditional distribution, we propose a conditional generalization of the infilling objective used to train T5 (Raffel et al., 2019) and ILM (Donahue et al., 2020). This process consists of several steps, also illustrated in Figure 1:

1. Replacing random spans of the tokenized IUPAC name with sentinel tokens.

2. Prepending the resulting sequence with a token indicating the original molecule's computed property value. To obtain these property value tokens, we discretize the distribution of property values into three buckets, specified in Table 5.

3. Training the model as in T5 to produce the sequence of redacted tokens, prepended by their corresponding sentinel tokens.

This conditional infilling objective incentivizes the model to learn the relationship between the computed property value and the missing tokens. To make a localized edit to a molecule, we then replace the desired fragments with sentinel tokens, change the property value token, and sample autoregressively from the predictive distribution. Thus, our approach hybridizes the flexible editing capabilities of ILM (Donahue et al., 2020) with the controllability of CTRL (Keskar et al., 2019). See Appendix C for experimental details. Code is available at `redacted`.

## 4 RESULTS

To demonstrate the promise of combining IUPAC names with conditional modeling using T5, we explore several molecular optimization tasks relevant to drug discovery. Specifically, we train C5T5 to make localized changes that affect the octanol-water partition and distribution coefficients (logP, logD), polar surface area (PSA), and refractivity—four properties commonly used to estimate bioavailability of a candidate drug (Ghose et al., 1999; Veber et al., 2002; Bhal et al., 2007). logP and logD measure the ratio of a compound's solubility in octanol, a lipid-like structure, and water; drugs need to be somewhat soluble in both to be orally absorbed. PSA and refractivity both relate to charge separation within the molecule. As shown in Appendix B, C5T5 generates mostly valid and novel molecules, with values of logP that lie outside of the range of a "best in dataset" baseline for targeted modifications.

### 4.1 C5T5 SUCCESSFULLY MODIFIES PROPERTIES

First, we demonstrate that the localized changes proposed by C5T5 do in fact control the property value as desired. C5T5 allows domain experts to choose where to make changes to a molecule based on their intuition and particular application: the user masks out tokens in the IUPAC name that can be modified, and the model fills in the masked spans in a way that changes the property value as directed. There is no canonical way to choose particular tokens to mask for evaluation purposes, so we simply choose a number of starting molecules randomly from PubChem (Kim et al., 2016), and then we iteratively mask all length-one to length-five spans (to match the training distribution) and run inference. We also experiment with masking multiple spans per molecule during inference, as is done during training. This is useful in practice when there are multiple areas of the molecule that can be changed in tandem to achieve the desired property value, but in our evaluation we observe qualitatively similar results when masking only a single span, so for computational efficiency we limit ourselves to single spans. We expect multi-span masks to be much more important when optimizing for more complex properties.

Figure 4 shows that C5T5 successfully generates molecules with higher property values when passed `<high>`, and with lower property values when passed `<low>`. The model is much more successful at raising property values than lowering them, especially for refractivity and polar surface area. For both of these properties, increasing the property value is straightforward: just add polar groups to replace whatever tokens were masked. In contrast, lowering these properties is only possible when the mask coincides with a polar group, in which case the model must find a non-polar substitute while still maintaining the molecule's validity. Even if unsuccessful at lowering these two property values on average, C5T5 can still be used in this case to suggest a number of candidate edits, and the one with the lowest property value can be selected using a property prediction model. This is an improvement over high-throughput screening and untargeted machine-learning methods for

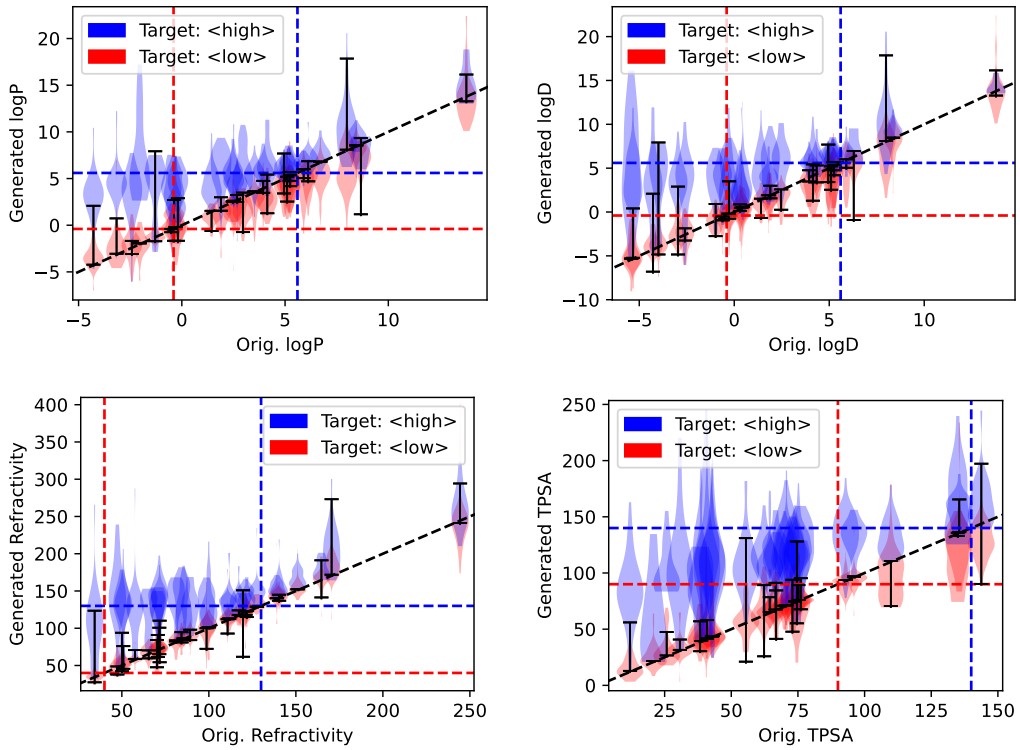

Figure 4: **Calculated property values of optimized molecules vs. original.** Values computed for 30 randomly chosen starting molecules. Top left: octanol-water partition coefficient. Top right: octanol-water distribution coefficient at pH of 7. Bottom left: molar refractivity. Bottom right: polar surface area. Blue violins show the distribution of generated molecule properties when the model was asked to complete molecules to achieve a `<high>` property value, and red for `<low>`. Cutoffs between `<high>` and `<med>` and between `<med>` and `<low>` are shown as dashed blue and red lines, respectively. The black dashed line is y=x. Black vertical lines show best in dataset baseline for each trial molecule.

molecular optimization, since it isn't restricted to a predefined library of candidate molecules, and it still allows the user to choose particular parts of the molecule to modify.

## 4.2 MODIFIED TOKENS ARE CHEMICALLY INTUITIVE

One main advantage of C5T5 is that suggested edits are in the intuitive language of IUPAC names, rather than arbitrary combinations of atoms. Table 1 shows the tokens that the model most preferentially adds to a molecule when asked to produce low vs. high logP. Unsurprisingly, the most common tokens added when increasing logP are generally long carbon chains (pentadecyl, undecyl, heptadecyl) and other hydrocarbons (trityl, perylen). Conversely, when the model is asked to produce low-logP modifications, hydrophilic groups are added. LogP is a simple metric, and by proposing molecular edits for logP that are obvious and easily understandable to domain experts, we expect users to gain confidence that C5T5 will suggest reasonable edits for more complex properties.

To further investigate the types of optimizations C5T5 suggests, Figure 5 visualizes two of the starting molecules from Figure 4 with logP values of $-2.4$ and $5.8$. For each molecule, we mask spans as usual and generate after prepending with `<low>` (molecules on the left) and `<high>` (molecules on the right). The IUPAC name of the top starting molecule is "3,3-bis(aminomethyl)pentane-1,5-diol," where "bis" signifies that the CNH2 group should be duplicated, and "diol" means duplicate OH groups at the ends. By virtue of the IUPAC name encoding symmetry, C5T5 is easily able to generate similarly symmetric molecules. For example, the molecule in the top left is "3,3-bis(aminomethyl)pentane-1,5-disulfinic acid," where the "ol" has been replaced with "sulfinic acid."

Table 1: **Tokens most preferentially added when C5T5 is asked to make modifications resulting high vs. low logP values.** Multipliers compare the actual rate of adding tokens compared to the expected number if the model drew randomly from the data distribution independent of property value. Blue tokens are hydrocarbons (i.e. lipophilic groups). Red tokens contain hydrogen bonding donors or acceptors (i.e. hydrophilic groups)

| Target: <high> | | Target: <low> | |
|---|---|---|---|
| trityl | 77.0x | phospho | 48.1x |
| pentadecyl | 20.2x | phosphonato | 44.7x |
| Z | 17.7x | sulfinam | 41.2x |
| perylen | 11.8x | hydrazon | 34.3x |
| undecyl | 8.1x | sulfinato | 26.3x |
| heptadecyl | 7.9x | Z | 17.9x |
| ylium | 7.6x | oxonium | 10.3x |
| isoindolo | 5.9x | amoyl | 9.3x |
| bH | 5.9x | carbamic acid | 8.6x |
| iod | 5.8x | sulfin | 6.9x |

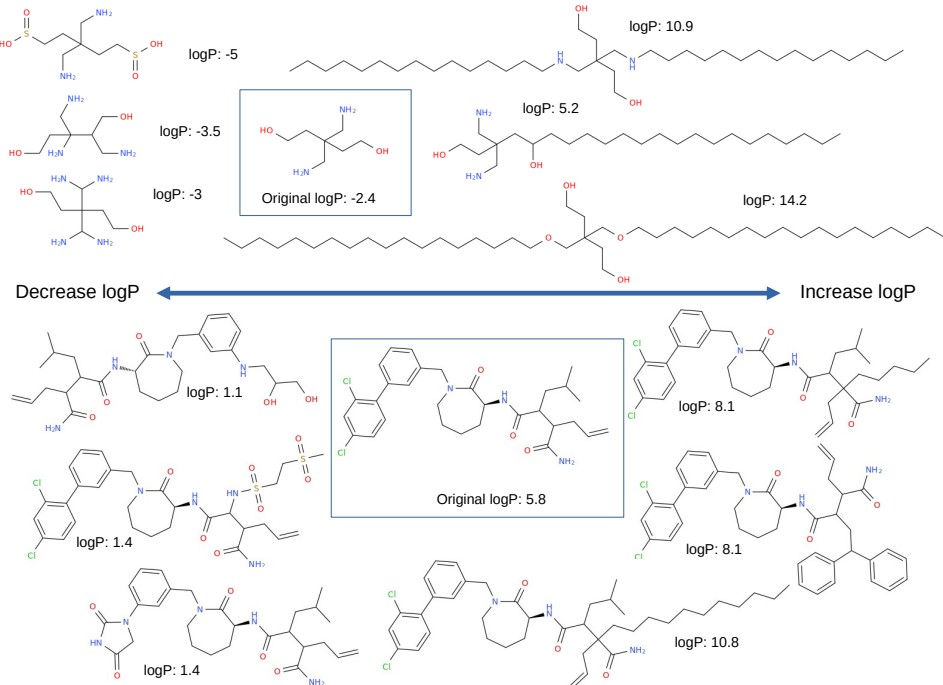

Figure 5: **Visualizations of base and logP-optimized molecules**. Two molecules from the logP plot in Figure 4, with three C5T5-optimized molecules for each of <low> and <high> logP.

Although symmetry is often highly desired, C5T5 is not limited to generating symmetric molecules. For example, the middle molecule on the right is "3,3-bis(aminomethyl)-1-heptadecylpentane-1,5-diol"—C5T5 added an additional carbon chain non-symmetrically at the end of the pentane.[3]

Sometimes C5T5 generates valid but unstable compounds. For example, the neighboring $NH_2$ groups in the bottom left molecule in the top half of Figure 5 are unstable, and would turn into aldehydes in an aqueous solution. All machine learning methods are susceptible to this sort of mistake, underscoring the importance of the type of human-in-the-loop optimization that C5T5 enables.

---

[3]Although this is not a preferred IUPAC name, it is still unambiguous, and therefore valid and parseable.

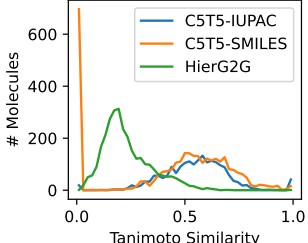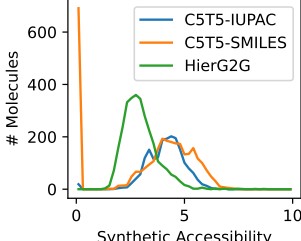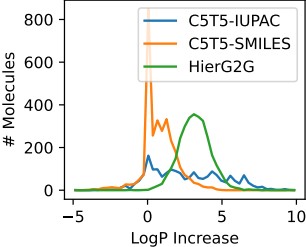

Figure 6: Histograms of Tanimoto similarity to base molecules, synthetic accessibility of generated molecules (both as computed in (Jin et al., 2020)), and logP increase for 30 random base molecules from the Zinc test set.

### 4.3 C5T5 MAKES SMALL AND SYNTHETICALLY ACCESSIBLE EDITS

C5T5's proposed molecular edits are not only intuitive, but also small and synthetically accessible. Unlike methods that train on edited pairs of molecules or that optimize molecules to be similar and synthetically accessible in a latent space, C5T5 achieves this purely from self-supervision.

For 30 base molecules randomly chosen from the Zinc dataset (Sterling & Irwin, 2015), we compute the Tanimoto similarity between generated and base molecules and the synthetic accessibility of generated molecules. Results are shown in Figure 6, along with a comparison to Hierarchical Graph-to-graph VAE (HierG2G), a state-of-the-art molecular graph translation model that trains on pairs of Zinc molecules (Jin et al., 2020). We also compare to an ablation of C5T5 that uses SMILES instead of IUPAC names. C5T5 generates molecules that are more synthetically accessible and similar to base molecules and that have a wider range of logP increases than HierG2G. An additional advantage of C5T5 not shown in Figure 6 is that C5T5 allows the user to choose what part of the molecule to edit, whereas HierG2G makes edits wherever it sees fit.

Unlike C5T5, methods like HierG2G that train on pairs of similar molecules are fundamentally limited by a paucity of experimental molecule pairs that have high similarity and high property value improvement. See Appendix B.2 for a quantitative analysis.

### 5 DISCUSSION AND CONCLUSION

We propose C5T5, a simple and effective zero-shot method for targeted control of molecular properties with transformers. Unlike prior approaches that make user-targeted modifications, our method requires no database of paired edits; instead, it simply trains in a self-supervised fashion on a large dataset of molecules and coarse estimates of their molecular property values. Core to our method is the use of IUPAC names as a data representation, which captures molecular structure at an appropriate level of abstraction and enables an intuitive editing interface for domain experts. C5T5 successfully rediscovers chemically-intuitive strategies for altering four drug-related properties in molecules, a notable feat given the absence of any human demonstration of these editing strategies.

Our work also has several limitations. The select-and-replace interface provided by the infilling objective may not always match the needs or preferred design process of domain experts. The interface also only suggests how to fill in missing parts of a molecule, relying on domain expertise or enumeration to decide which parts of the molecule should be changed to begin with. In addition, we explore only a coarse-grained bucketing of property values, leaving a more fine-grained treatment for future work. IUPAC names might also be too limiting in cases where a user wants to edit a subgraph of a molecule that does not correspond neatly to a small number of IUPAC tokens. Finally, training C5T5 is computationally expensive and sample inefficient.

Future work will investigate using C5T5 with more molecular properties, such as the power conversion efficiency of solar cells. We also leave to future work extending C5T5 to jointly model multiple properties, and adding a more flexible editing interface.

## 6 REPRODUCIBILITY

In Appendix C, we describe how to download and prepare the training dataset, our tokenization strategy, masking strategy, optimizer, learning rate schedule, number of iterations, and generation strategy. Our code is also available online at REDACTED.

One barrier to reproducibility is that we use ChemAxon's proprietary software to compute molecular property values. However, ChemAxon offers a free academic license, and we suspect that all results would be similar if property values were computed with RDKit instead.

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

| Method | Base Repr. | Model | T?[a] | UD?[b] | CO?[c] |
|---|---|---|---|---|---|
| C5T5 | IUPAC | T5 | ✓ | ✓ | ✓ |
| He et al. (2021b) | SMILES | Seq2Seq/Transformer | ✗ | ✗ | ✓ |
| Langevin et al. (2020) | SMILES | Any | ✓ | ✗ | ✗ |
| Arús-Pous et al. (2020) | SMILES | LSTM | ✓ | ✗ | ✗ |
| Jin et al. (2019) | Graph/Motifs | JT-VAE+GAN | ✗ | ✗ | ✗ |
| Li et al. (2019) | Graph/Atoms | GNN+VAE | ✗[d] | ✗ | ✗ |
| Lim et al. (2020) | Graph/Atoms | VAE+GNN | ✗ | ✗ | ✓ |
| Maziarz et al. (2021) | Graph/Motifs | VAE+GNN | ✗ | ✓ | ✓ |
| Bagal et al. (2021) | SMILES | GPT | ✗ | ✓ | ✓ |
| Shin et al. (2021) | SMILES | Transformer+LSTM | ✗ | ✗ | ✓ |
| He et al. (2021a) | SMILES | Transformer | ✓ | ✗ | ✓ |
| Kotsias et al. (2020) | SMILES | cRNN | ✗ | ✓ | ✓ |
| Gómez-Bombarelli et al. (2018) | SMILES | VAE+ConvNet+GRU | ✗ | ✓ | ✗ |
| Lim et al. (2018) | SMILES | cVAE | ✗ | ✓ | ✓ |
| Dollar et al. (2021) | SMILES | VAE+Transformer | ✗ | ✓ | ✗ |
| Liu et al. (2018) | Graph/Atoms | VAE+GNN | ✗ | ✓ | ✗ |
| You et al. (2018) | Graph/Atoms | GCN+GAN+RL | ✗ | ✓ | ✗ |
| Jin et al. (2018) | Graph/Motifs | JT-VAE | ✗ | ✓ | ✗ |
| Maziarka et al. (2020) | Graph/Motifs | CycleGAN+JT-VAE | ✗ | ✓ | ✓ |
| Shi* et al. (2020) | Graph/Atoms | Flow+RL | ✗ | ✓ | ✗ |
| Olivecrona et al. (2017) | SMILES | RNN+RL | ✗ | ✓ | ✗ |

Table 2: Comparison of C5T5 to prior methods

[a]Whether or not the model makes targeted modifications – *i.e.* whether the user can specify which part of the molecule the model should modify.

[b]Whether or not the model can train without using paired molecular data.

[c]Whether or not the model can switch objectives without re-training or re-optimizing.

[d]The authors propose filtering results based on where the user wants to add a side chain, but the method itself does not target specific attachment points.

## A  QUALITATIVE COMPARISON TO PRIOR METHODS

Table 2 shows how C5T5 and prior methods for molecular optimization differ along several axes.

## B  ADDITIONAL EXPERIMENTS

### B.1  NOVELTY, VALIDITY, & BEST-IN-DATASET

Section 4.1 shows that C5T5 successfully modifies property values. Here, for molecules generated to optimize logP, we show the novelty and validity of the generated molecules, along with a comparison to a baseline of the best eligible compound in PubChem. To match how the model was trained and how new molecules were generated, eligible compounds are those that could be generated by masking any consecutive span of at most 5 IUPAC name tokens and replacing the masked tokens with any number of replacement tokens.[4] Results are shown in Tables 3 and 4. Percent validity is the fraction of generated molecules that T5 generated with valid sentinel tokens that were considered chemically valid by the ChemAxon logP calculator. Percent novelty is the fraction of distinct generated molecules that do not appear in PubChem (excluding when C5T5 re-generated the source molecule). As shown, despite the comparative difficulty of learning IUPAC name syntax compared to SMILES syntax, C5T5 consistently finds novel and valid molecules that significantly outperform the best-in-dataset baseline.

[4]For computational efficiency, we filter out molecules that differ in length by more than 15 tokens, or that have more than 15 non-overlapping tokens in their bag of tokens, before checking whether they could indeed be generated by masking some length-5 sequence of tokens.

Table 3: For each source molecule, we show the percent of generated molecules that are novel (not in PubChem, including invalid names) and valid (can be parsed by ChemAxon), the number of generated molecules, the min/max logP of any generated molecule, the number of eligible compounds in PubChem and the min/max logP of all molecules in PubChem that could be generated by masking up to 5 consecutive tokens. IUPAC names of source molecules are listed in Table 4.

| Src. | # gen. | % novel | % valid | max gen. | min gen. | # elig. | max PC | min PC |
|------|--------|---------|---------|----------|----------|---------|--------|--------|
| 1  | 82  | 95.1%  | 81.7% | 14.22 | -5.04 | 26  | 10.81 | -2.7  |
| 2  | 133 | 93.2%  | 91.7% | 9.41  | -3.82 | 28  | 1.92  | -1.46 |
| 3  | 217 | 98.6%  | 91.7% | 9.15  | -2.17 | 4   | 3.01  | 1.66  |
| 4  | 140 | 100.0% | 94.3% | 10.21 | 1.15  | 3   | 8.18  | 3.78  |
| 5  | 128 | 92.2%  | 88.3% | 6.91  | 0.86  | 19  | 4.24  | 2.35  |
| 6  | 160 | 100.0% | 75.6% | 8.16  | 0.47  | 4   | 2.56  | 1.6   |
| 7  | 159 | 99.4%  | 86.2% | 9.05  | -2.22 | 8   | 3.91  | -1.34 |
| 8  | 137 | 99.3%  | 84.7% | 14.74 | -2.97 | 1   | 1.08  | 1.08  |
| 9  | 122 | 95.1%  | 81.1% | 8.57  | -4.89 | 34  | 4.75  | -0.07 |
| 10 | 112 | 92.9%  | 88.4% | 8.44  | -2.96 | 112 | 5.03  | -2.3  |
| 11 | 127 | 97.6%  | 81.1% | 9.52  | 0.11  | 9   | 5.13  | 2.28  |
| 12 | 114 | 90.4%  | 85.1% | 7.71  | -3.32 | 515 | 6.13  | -2.95 |
| 13 | 115 | 94.8%  | 90.4% | 8.12  | -1.91 | 36  | 4.05  | 0.3   |
| 14 | 149 | 97.3%  | 85.2% | 14.09 | -4.3  | 7   | 2.1   | 0.44  |
| 15 | 135 | 100.0% | 83.0% | 7.52  | -0.7  | 2   | 5.08  | 4.12  |
| 16 | 214 | 100.0% | 97.7% | 6.86  | -1.83 | 3   | 1.91  | -0.05 |
| 17 | 156 | 99.4%  | 92.3% | 10.58 | -2.88 | 2   | 0.76  | 0.76  |
| 18 | 231 | 98.7%  | 83.1% | 9.79  | -1.35 | 6   | 5.6   | 3.46  |
| 19 | 148 | 93.2%  | 82.4% | 9.98  | -0.19 | 63  | 7.01  | 0.76  |
| 20 | 143 | 96.5%  | 72.7% | 14.12 | 0.33  | 14  | 4.61  | 1.31  |
| 21 | 232 | 99.1%  | 85.3% | 9.9   | -1.6  | 6   | 3.15  | 1.14  |
| 22 | 150 | 96.0%  | 92.7% | 7.96  | -0.74 | 22  | 4.95  | 2.71  |
| 23 | 127 | 94.5%  | 87.4% | 7.54  | -3.1  | 18  | 4.67  | 0.37  |
| 24 | 160 | 99.4%  | 82.5% | 7.54  | -1.53 | 2   | 1.28  | 1.28  |
| 25 | 274 | 100.0% | 95.6% | 10.82 | 1.12  | 2   | 6.12  | 5.82  |

## B.2 EDITED MOLECULE PAIRS IN ZINC

Methods like HierG2G (Jin et al., 2020) train a model to translate between "unoptimized" and "optimized" molecules, where the training set consists of pairs of similar molecules in an underlying dataset (e.g. Zinc) that have different property values.

Training a model in this way faces the fundamental limitation that there are very few pairs of molecules that are both similar (e.g. in Tanimoto similarity space) and have very different property values. To quantify this effect, we compute similarity and logP increase between all pairs of molecules in the training set of Zinc used by Jin et al. (2020). Results are shown in Figure 7.

## C EXPERIMENTAL DETAILS

**Dataset Preparation**   We download PubChem from `ftp.ncbi.nlm.nih.gov/pubchem/Compound/CURRENT-Full/XML/` and extracted each molecule's Preferred IUPAC Name and computed octanol-water partition coefficient (logP). There are 109M total compounds in the version of PubChem we downloaded in January 2021. For experiments using logP, we used the XLogP3 values from PubChem, which were computed using OpenEye's software. For logD (pH=7), refractivity, and polar surface area, we computed values using ChemAxon's calculator with default parameters. Separately for each target property, we excluded chemicals that had no logP value in PubChem or that were not parsable by ChemAxon's calculator. Of the remaining molecules, we randomly split into a training set with 90M compounds and a validation set with $\sim$10M-19M compounds.

Table 4: IUPAC Name lookup table for Table 3

| ID | Source Molecule |
|----|-----------------|
| 1  | 3,3-bis(aminomethyl)pentane-1,5-diol |
| 2  | 1-(3-hydroxypropyl)-N-(1-methoxybutan-2-yl)pyrazole-4-sulfonamide |
| 3  | 4-chloro-N-[2-[[2-(4-fluorophenyl)acetyl]amino]ethyl]-1,3-thiazole-5-carboxamide |
| 4  | 1-[(1S)-1-(3-fluorophenyl)propyl]-3-iodoindole |
| 5  | 4-(4-fluorophenyl)-N-[(1R,2R)-2-methylcyclohexyl]piperazine-1-carbothioamide |
| 6  | N'-(3-ethyl-4-oxophthalazine-1-carbonyl)-4-methyl-2-phenyl-1,3-thiazole-5-carbohydrazide |
| 7  | (E)-2-methoxy-3-methylhex-4-en-1-ol |
| 8  | N-methyl-1-[2-(4-methylthiadiazol-5-yl)-1,3-thiazol-4-yl]methanamine |
| 9  | 2-[(7-methyl-[1,2,4]triazolo[1,5-a]pyridin-2-yl)amino]ethylurea |
| 10 | 4-[[2-(2-oxopyridin-1-yl)acetyl]amino]benzoic acid |
| 11 | [6-prop-2-enoxy-4-(trifluoromethyl)pyridin-2-yl]hydrazine |
| 12 | 4-(2-methylphenyl)sulfonylpiperidin-3-amine |
| 13 | 3-[ethyl(2-methylpropyl)amino]propane-1-thiol |
| 14 | 6-methoxy-4-N-methyl-4-N-[(2-methylfuran-3-yl)methyl]pyrimidine-4,5-diamine |
| 15 | 3-phenylmethoxy-5-(trifluoromethoxy)quinoline-2-carboxylic acid |
| 16 | 3-[4-[acetamido-[3-methoxy-4-[(2-methylphenyl)carbamoylamino]phenyl]methyl]piperidin-1-yl]-3-phenylpropanoic acid |
| 17 | (3R)-3-[[(2S)-2-[benzyl(methyl)amino]butanoyl]amino]pyrrolidine-1-carboxamide |
| 18 | 6-cyclobutyl-2-N-[3-(1-ethylsulfinylethyl)phenyl]-5-(trifluoromethyl)pyrimidine-2,4-diamine |
| 19 | 6-fluoro-2-(4-phenylpyridin-2-yl)-1H-benzimidazole |
| 20 | 4-chloro-3-(2-oxo-1,3-dihydroindol-5-yl)benzonitrile |
| 21 | 1-(6-tert-butylpyridazin-3-yl)-N-methyl-N-[(2-methyl-1,3-oxazol-4-yl)methyl]azetidin-3-amine |
| 22 | 2-[(4aR,8aS)-3,4,4a,5,6,7,8,8a-octahydro-1H-isoquinolin-2-yl]-N-(2,4-dimethoxyphenyl)acetamide |
| 23 | 2-[2-(2,4-dichlorophenoxy)ethoxy]-4-methoxybenzoic acid |
| 24 | (2Z)-2-[(1,7-dimethylquinolin-1-ium-2-yl)methylidene]-1-ethyl-7-methylquinoline |
| 25 | N'-[(3S)-1-[[3-(2,4-dichlorophenyl)phenyl]methyl]-2-oxoazepan-3-yl]-3-(2-methylpropyl)-2-prop-2-enylbutanediamide |

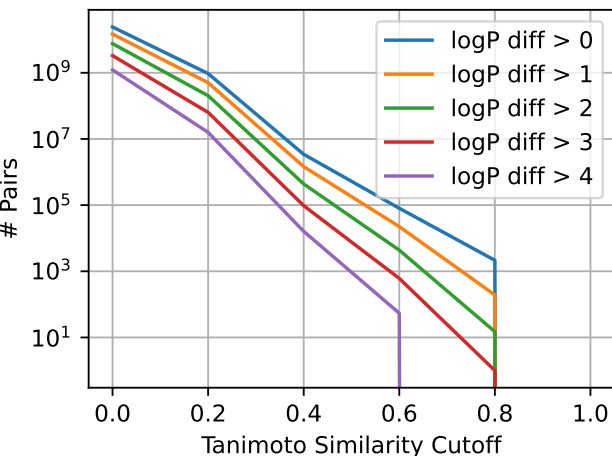

Figure 7: Number of molecular pairs in Zinc that pass a Tanimoto similarity threshold (x-axis) and a logP-difference threshold (different curves). Even for moderate similarity and logP thresholds, there quickly become vanishingly few usable molecular pairs. There are $24 \times 10^9$ total unique molecular pairs.

**Tokenization**   We use HuggingFace's T5Tokenizer, which is based on the SentencePiece algorithm Kudo & Richardson (2018). Because our goal is to have tokens that domain experts are familiar with, we do not train SentencePiece on the IUPAC names, since doing so learns both combinations of and substrings of moiety names. Instead, we manually specify the tokens to be all the keywords in the Opsin IUPAC name parsing library Lowe et al. (2011). To these keywords, we add locants 1–100,

stereochemistry markers (R, S, E, Z,...), and a few miscellaneous tokens to e.g. handle spiro centers. This leads to a total of 1274 tokens. After tokenization, we truncate all names to at most 128 tokens.

We choose which property value token to use based on the cutoffs specified in Table 5.

Table 5: **Numerical ranges across properties for each property value token.** Cutoffs for logP, PSA, and refractivity were chosen as common thresholds for druglikeness screening following (Ghose et al., 1999; Veber et al., 2002; Hitchcock & Pennington, 2006). We use the same cutoff for logD as logP.

| Property | `<low>` | `<med>` | `<high>` |
|---|---|---|---|
| Octanol-water partition coeff. (logP) | $(-\infty, -0.4)$ | $(-0.4, 5.6)$ | $(5.6, \infty)$ |
| Octanol-water distribution coeff. (logD) | $(-\infty, -0.4)$ | $(-0.4, 5.6)$ | $(5.6, \infty)$ |
| Polar surface area (PSA) | $(0, 90)$ | $(90, 140)$ | $(140, \infty)$ |
| Refractivity | $(0, 40)$ | $(40, 130)$ | $(130, \infty)$ |

**Training**   We train a t5-large model ($\sim$700M params) available from HuggingFace that was pre-trained on English text. We keep the first 1274 embeddings from the pretrained embedding table, along with the pretrained embeddings for the 100 sentinel tokens. When training, we mask 15% of the tokens in each input in spans of mean length 3 tokens, with a minimum span length of 1. We use a linear warmup of the learning rate for 10,000 steps followed by a $1/T$ decay. All models were trained using the AdamW optimizer. We train the logP model for 2.5M iterations using a max learning rate of $10^{-3}$. We train the refractivity/logD/Polar SA models using a maximum learning rate of $2 \times 10^{-4}/10^{-4}/2 \times 10^{-4}$ starting from the logP model after 1M/2.5M/2.5M iterations. (We trained using the latest model available when we started a run.)  All models were trained on 8 NVIDIA A100s with a batch size of 16 per GPU.

**Generation**   We generate novel molecules greedily from the output of T5's decoder. We discard any generations where the sentinel tokens do not line up, and we further discard any molecules that cannot be parsed by ChemAxon's calculators. We also discard instances where C5T5 regenerates the base molecule. Doing so inflates our novelty score, but because we choose random masks (including, e.g. masking out a comma between two locants), in many cases there are few or no other ways to fill in the masked section besides regenerating the original tokens.

**Comparison to HierG2G**   We trained HierG2G on Zinc using the authors' code and hyperparameters from (Jin et al., 2020). We generated all pairs of Zinc training molecules with a Tanimoto similarity of at least 0.4 and an increase in logP of at least 2.5, yielding $\sim$200k pairs. We trained HierG2G for 32k iterations; further training did not improve results. To choose the similarity and logp thresholds, we attempted to use the average similarity and logp increase achieved by C5T5 (similarity of 0.55, logP increase of 2.4). However, there are only 7000 pairs of molecules in Zinc meeting these constraints, so we relaxed the cutoffs to achieve a reasonably sized training set. See Appendix B.2 for further details.

We were not able to fully reproduce the results in (Jin et al., 2020), either by running the authors' code as-is or by modifying the hyperparameters to match the description in the paper. Specifically, for penalized logP with a similarity cutoff of 0.4, we get an average improvement of $\sim 2.8$, whereas the results in the paper show an improvement of 3.98. We will update Figure 6 once we are able to reproduce the paper's results, but we do not believe the conclusions of our comparison will change qualitatively even with updated results from HierG2G.

**Cloud Computing Cost**   We train T5-Large models Raffel et al. (2019) on PubChem for each investigated property value using local computing resources and AWS p4d.24xlarge instances in the us-east-1 region. The logP model was trained for 2.5M iterations over 8 days; the logD and refractivity models for 350k iterations over 1 day each (initialized from the logP model after 2.5M iterations); and the Polar SA model for 1.6M iterations over 5 days (initialized from the logP model after 1M iterations). The SMILES logP ablation model was trained for 1M iterations over 12 days.

Total equivalent on-demand AWS cost for the models presented here is $\sim$ \$19,000, and total carbon dioxide-equivalent emissions is $\sim$ 570 kg Lacoste et al. (2019).

**Software** We use HuggingFace 4.2.2 (Apache 2.0 license) (Wolf et al., 2019), PyTorch 1.8.0 (BSD) (Paszke et al., 2019), ChemAxon 20.17.0 (Academic License), scikit-learn 0.24.2 (New BSD) (Pedregosa et al., 2011) and python 3.9 (PSFL).

