# OpenReview forum: "C5T5: Controllable Generation of Organic Molecules with Transformers"
_ICLR.cc/2022/Conference — ICLR 2022 Submitted_

### Official Review · Reviewer_MJUv · 2021-11-02

**Correctness:** 3
**Technical Novelty And Significance:** 3
**Empirical Novelty And Significance:** 2
**Recommendation:** 5
**Confidence:** 4

**Main Review:**

The summary of the paper is concluded in the above contents. Here are the strengths and weaknesses of the paper from my view.

Strengths:
1. The first attractive point is the IUPAC names representation method. As mentioned in the paper, this method is a standardized molecular representation that encodes the rich structural information but that has been largely ignored. The common approach of the representation for the molecule is the SMILES or graph. As introduced, this representation can handle more semantic information, and also the interpretability is better.
2. The method is straightforward and simple, which uses a pre-training method T5 to model a conditional language model, where the condition is the property value. As a result, C5T5 does not require paired dataset for the molecule property improvement.
3. The results on properties show the desired optimization is satisfied, and the different studies give a more clear understanding.

weaknesses:
Though I personally like the simple idea, there are still concerns and questions from my point,
1. The first concern is about the specific property value distribution. As shown in Table 5, each property only splits into three buckets. Then each model can only have three different types of property, which makes the model to be limited. For example, it can not clearly generate a better molecule if the initial molecule with logP is larger than 5.6.
2. The second concern is about the IUPAC names. Though I feel happy to see a new representation method that seems to be better than other methods, the comparison is not clear and there seems no strong evidence or numbers can be shown in the paper. In this way, this can not clearly convince the superiority of IUPAC. However, I am pretty interested to see the goodness of IUPAC.
3. One specific question is about the pre-trained T5 model. The authors mentioned that they do not use sentencepiece, and only keep the IUPAC names as the vocabulary, which is different from the pre-trained embedding of T5 model. It is said that the first 1274 embeddings of English pre-trained embedding are kept. I am not sure about this process, what is the difference between this method and directly replacing the vocabulary with the 1274 new randomly initialized embeddings? Since the vocabulary is different at all.
4. It is required to make a result comparison to see the advantage of this method. Otherwise, it is hard to evaluate.
5. Other concerns have been discussed by the authors, for example, the training cost is high; it is not always possible to generate valid and satisfying molecules since there are no constraints in the modeling.



**Summary Of The Paper:**

This paper proposes a method C5T5, a self-supervised pre-training method based on the T5 pre-trained model, which is able to make zero-shot select-and-replace edits to satisfy specific property values. The specific difference of this paper is the IUPAC names (a standardized molecular representation), and the method is totally self-supervised. The experiments are evaluated on octanol-water partition, distribution coefficients, polar surface area, and refractivity, above four properties. Experiments show that the designed methods are able to achieve the optimization objective.

**Summary Of The Review:**

I personally feel good about this work, but unclear parts are main concerns and there are no comparison with other works.

---

> ### Author Response · Authors · 2021-11-20
> **Response to R3**
>
> * “each property only splits into three buckets”
>
> We are targeting the task of optimizing lead molecules to improve their ADMET characteristics. In this case, practitioners generally want properties to fall within some reasonable range, or to be at least or at most some value. We therefore don’t see our binning of property values as a significant limitation. In cases where finer control over property values is desired, we can simply use more than 3 property bins.
> * “In this way, this can not clearly convince the superiority of IUPAC.”
>
> See overall comment
> * “what is the difference between this method and directly replacing the vocabulary with the 1274 new randomly initialized embeddings?”
>
> We found that training goes faster if we initialize the model (including token embeddings) with the weights from a model pretrained on English. As you point out, the IUPAC vocabulary is very different from English. Prior work has found that to some extent, pretrained transformers can generalize to radically different tasks and modalities (e.g. text to music or code arxiv:2004.14601), and we find similar results here as well.
> * “It is required to make a result comparison”
>
> See overall comment
> * “the training cost is high”
>
> Preliminary results suggest that we can train for much less time and achieve similar quality of results. However, we have not yet engaged in a serious effort to reduce the computational cost of C5T5. Our primary goal has been to demonstrate that the method can be effective, leaving computational efficiency to future work. Future work can also explore finetuning an unconditional model on many different properties, or leveraging pretrained models for other tasks, allowing for reuse and more computational efficiency.
> * “it is not always possible to generate valid and satisfying molecules since there are no constraints in the modeling.”
>
> Adding constraints to IUPAC generation is much more difficult than adding constraints to SMILES or graph generation, since the IUPAC grammar is complex. Nonetheless, C5T5 mostly generates valid molecules, as shown in Table 3.

---

> > ### Comment · Reviewer_MJUv · 2021-11-23
> > **It's still questioned**
> >
> > Though I feel happy at the first glance, now I am a little confused and not so convinced about the rebuttal.
> >
> > The authors replied that IUPAC can not compare with SMILES or graphs due to different masking strategies or functional group masking. Indeed, masking strategy can be defined manually, such as BPE mask or whole-word-BPE mask.
> >
> > It is also confusing that the experiments can not be compared because the tasks are different. I think that IUPAC is not specially designed for some specific tasks.
> >
> > As for the initialized embedding, do you just directly copy the first 1274 tokens to remap to the IUPAC vocabulary? Is it right? Just reuse the embedding?

---

> > > ### Author Response · Authors · 2021-11-26
> > > **Response to R3**
> > >
> > > Thanks for following up! Please let us know if this response addresses your questions. We are happy to follow up more:
> > >
> > > **Comparability of SMILES to IUPAC names**
> > >
> > >
> > > It's certainly true that one can define masking strategies manually, e.g. using whole-word masking instead of BPE masking! However, the relationship between SMILES and IUPAC is much more complex than the relationship between BPE and whole words in natural language. For example, there is no parallel in SMILES to masking a multiplier token, a token that has been multiplied, a locant token, etc. Thus there is no manual masking strategy that enables a mapping between the two. This flexibility is part of what makes IUPAC such a powerful and interpretable interface.
> > >
> > > **Initialization of token embeddings**
> > >
> > > You are correct—for the initial embeddings, we directly copy the first 1274 tokens from an English-pretrained model. The embeddings are then trained along with the rest of the model.
> > >
> > > **Why we focus on different problems from past work**
> > >
> > > C5T5 addresses a task we call select-and-replace editing, where the domain expert chooses a part of the molecule to modify, and the model suggests changes to that part of the molecule. This is different from previous work on finding an optimal molecule subject to the constraint that the molecule must contain a specific subgraph. In the latter case, the model can append to the subgraph anywhere it wants, which is a much less constrained problem than select-and-replace editing. We anticipate the select-and-replace task to be much more useful for domain experts who are optimizing drugs for ADMET properties later on in the drug discovery pipeline, when a lead molecule has already been identified and characterized to some extent. In that case, medicinal chemists already have some understanding of the molecule’s structure-activity relationship, and are likely to need more control over the generative process as a result.
> > >
> > > With respect to masking IUPAC tokens vs. e.g. BPE masking of SMILES, we see three advantages to IUPAC:
> > >
> > > 1. The first is that individual tokens are interpretable, unlike BPE tokens. The user needs to decide what to mask, and if the only available masks are seemingly random collections of SMILES tokens, the user will have a hard time using the system.
> > >
> > > 2. The second advantage is that IUPAC tokens are less likely to break up functional groups than SMILES, so there are most likely fewer possible representations for each moiety than there are with SMILES. Take as a simple example benzenediol. There are three possible arrangements of the alcohol groups: 1,2-benzenediol, 1,3-benzenediol, and 1,4-benzenediol (i.e. ortho, meta, and para). The ordering of carbons is arbitrary, and e.g. 2,3-benzenediol unambiguously refers to the same molecule as 1,2-benzenediol, even though it is not a preferred IUPAC name. However, no matter which locants are used, every IUPAC name contains “benzenediol”, which refers to the benzene ring, a 2x multiplier, and an alcohol group. In contrast, the benzene ring is represented in many different ways in SMILES notation. Even when using canonical SMILES, the three molecules are “C=1(C(=CC=CC1)O)O,” “C1(=CC(=CC=C1)O)O,” and “C1(=CC=C(C=C1)O)O,” each of which uses a different representation for benzene. When considering non-canonical SMILES, the situation becomes far worse. Using BPE on SMILES requires many different tokens just to refer to the one simple benzene ring.
> > >
> > > 3. The third advantage of IUPAC tokens over SMILES tokens is that IUPAC tokens support additional functionality like multiplier tokens (e.g. “di” in the example above) and locants. Editing a multiplier token allows for creating or eliminating symmetric groups, and editing a token that is multiplied allows easy editing of several symmetric groups at once. Editing a locant allows moving around attachment points of subgroups and rings. Neither of these are easy with SMILES edits.

---

### Official Review · Reviewer_gkwv · 2021-11-02

**Correctness:** 2
**Technical Novelty And Significance:** 3
**Empirical Novelty And Significance:** 2
**Recommendation:** 3
**Confidence:** 4

**Details Of Ethics Concerns:**

The authors forgot to include a statement.

**Main Review:**


Strenghts
The idea of using the IUPAC nomenclature is indeed innovative and has not received attention in the area. The use of local modification of tokenized IUPAC names is exciting and seems to work for scaffold derivatization better than other methods

The T5 model is a powerful addition to language-based addition to molecule generation

Weaknesses
At a higher level, this paper proposes a new generative approach but does not evaluate it upon the mutually agreed benchmarks in the field. Guacamol or Moses have lots of meaningful tasks and splits, including some scaffold-based ones. It does not seem appropriate to switch architectures and tasks and then just compare with the one baseline. If the authors believe those benchmarks are relevant, then it is more appropriate to make a case why explicitly (and probably use them anyway)

Furthermore, making local modifications to starting scaffolds has a long history in cheminformatics. Approaches like genetic algorithms, including the graph-based genetic algorithm, can do a great job at local optimization challenges like this https://pubs.rsc.org/en/content/articlelanding/2019/sc/c8sc05372c And since all the properties come from a surrogate model oracle, it does not matter how many train pairs or in-time calls need to be made. LogP (or logD, i think for all the molecules shown they are __extremely__ correlated) is an additive property that can be improved by just appending more carbons to an alkyl chain, so it is a poor performance metric. One could very easily make a genetic-algorithm version of the token substitution approach as a baseline.

I have a number of smaller comments below
- The advantages of using IUPAC names are not clear at all.

"a small change in a molecule can lead to a large change in the SMILES string" This is the case too for IUPAC names - taking one carbon away and closing a ring in the suggested structure in Figure 2 completely changes the IUPAC name

"flattening the graph into a list of atoms artificially creates variable- and long-range dependencies between bonded atoms" It does in the IUPAC naming too, since there's a canonical atom ordering that is needed to refer to for writing the molecular name

"It is difficult to reason about common substructures, because the same structure can be
represented in many different ways depending on how the graph was flattened"

Same in IUPAC nomenclature - the ester functionality is described through a completely different naming in the example below


OC(C1=C(OC(C)=O)C=CC=C1)=O
2-acetoxybenzoic acid

O=C(O1)C2=C(OC1=O)C=CC=C2
4H-benzo[d][1,3]dioxine-2,4-dione


"graphs do a poor job encoding symmetry, long-range interactions between atoms that are many bonds apart but nearby in 3D space, and long-range interactions that arise from conjugated system" Again, the IUPAC naming convention has many of these flaws too. There's no 3D whatsoever in IUPAC either.

"locants (e.g. “1,” “2,” “N”), which indicate connectivity," These locants are also based on arbitrary ordering, just like SMILES

Again just like SMILES, while there's only one canonical choice of IUPAC nomenclature based on prioritization rules, it is entirely possible to write the IUPAC name for a molecule correctly but based on another prioritization rule. As a matter of fact this paper's approach to tokenization likely does that, depending on the nature of the transformations it perform on the starting name.

Section 3.1 addresses few of the issues raised about SMILES / graphs. (for instance, it does address the multiplier, which is indeed very powerful in theory, but does not seem to be used by the generator in practice. "pentatrytilbenzene" would break the bank in logP)

"ignoring the embedding for “nitroso”" I would make this part of the sentence, rather than a footnote

"We discard any generations where the sentinel tokens do not line up, and we further discard any molecules that cannot be parsed by ChemAxon’s calculators. We also discard instances where C5T5 regenerates the base molecule" I think I understand the logic about needing to seed with something, but discarding the same from the statistics seems antithetic to the nature of a novelty metric. What is the fraction of molecules being discarded at every filter?

"Although this is not a preferred IUPAC name, it is still unambiguous, and therefore valid and parseable" This is exactly the point i was making earlier. It seems the approach struggles with synonyms just like SMILES.

"C5T5 generates molecules that are more synthetically accessible and similar to base molecules and that have a wider range of logP increases than HierG2G." I am not sure I'm looking at the plots right. How was synthetic accessibility computed? Is 0 high or low accessibility? The scale in RDKIT has 1 for easy molecule and 10 for difficult, So C5T5 is producing more difficult molecules. Same for the logP increase, i see HierG2G producing a mean of 4 or so, and C5T5 is a little over 1?



"Methods like HierG2G that train on pairs of similar molecules are fundamentally
limited by a paucity of experimental molecu pairs that have high similarity and high property
value improvement" This seems misleading since all these methods are using surrogate models to make their labels, there's essentially N squared pairs for the training library.



Questions

"The nearest neighbor of “diphosphate” - “disulfate” + “sulfate” is “phosphate." I don't follow. Are diphosphate and disulphate tokens? Does the tokenization procedure not split out the di? Or the "-ite" termination? Those are literally rule-based tokens in IUPAC nomenclature.

**Summary Of The Paper:**

This paper utilizes a transformer architecture based on T5 to generate tokenized IUPAC names of molecules. This allows sampling molecules that are local modifications of starting ones, or are local in IUPAC name but might be chemically large while respecting an initial scaffold.


**Summary Of The Review:**

I think the idea of using IUPAC names is intriguing, but neither the theoretical arguments not the empirical results (due to lack of benchmarks) are convincing.

---

> ### Author Response · Authors · 2021-11-20
> **Response to R2 [1/2]**
>
> * The use of local modification of tokenized IUPAC names is exciting and seems to work for scaffold derivatization better than other methods
>
> Thanks! We’d like to note that C5T5 could be used to decorate scaffolds, but nothing limits users to editing any particular part of the molecule -- any IUPAC token can be masked.
> * “this paper proposes a new generative approach but does not evaluate it upon the mutually agreed benchmarks in the field”
>
> See overall comment
> * “Approaches like genetic algorithms, including the graph-based genetic algorithm, can do a great job at local optimization challenges like this”
>
> See overall comment
> * “since all the properties come from a surrogate model oracle, it does not matter how many train pairs or in-time calls need to be made”
>
> See overall comment
> * “LogP...is an additive property that can be improved by just appending more carbons to an alkyl chain, so it is a poor performance metric”
>
> LogP may be a poor benchmark for methods like genetic algorithms that are allowed to go arbitrarily far out of the training distribution. For C5T5 we think it is a reasonable beginner benchmark, since the simple relationship between structure and logP allows us to build a case for the interpretability of our method as in Table 1. We also think it’s reasonable to start with such a simple benchmark since using IUPAC names for molecular modeling has never been previously explored. Unlike methods that optimize a molecule at inference time, C5T5 does not risk adding carbons ad infinitum to increase logP, since it learns to stay within the training distribution. Penalizing logP by synthetic accessibility score (as is commonly done in the literature) is therefore less important.
>
> * “One could very easily make a genetic-algorithm version of the token substitution approach as a baseline.”
>
> We can try this out and report back, but almost every randomly chosen combination of IUPAC tokens is invalid (initialization); choosing a random token to change almost certainly invalidates the name (mutation); and combining two different sets of IUPAC tokens is also very unlikely to lead to a valid, let alone improved, IUPAC string (children). It might be possible to manually specify which tokens could be substituted for which other tokens, but much of that is also context dependent. E.g. “2” can in principle be replaced by “99,” but only in the unlikely event that there are 99 numbered carbons.
> * “The advantages of using IUPAC names are not clear at all.
>
> Your points are well taken. We don’t intend to present IUPAC names as a cure-all for the difficulties encountered by other representations. However, we do think there is a case to be made that IUPAC names help alleviate some of these difficulties. For example, as you point out, common substructures can have multiple IUPAC representations, just as in SMILES, but the problem is much more severe in SMILES -- any common substructure that can be substituted ends up getting split up by the decorators, and has atoms arbitrarily ordered. Locants do depend on an arbitrary ordering, as in SMILES, but small edits to locant tokens allow moving around large parts of the molecule, which is not possible in SMILES (and is difficult for many graph methods). IUPAC names don’t include 3D information, but they do encode symmetry, and many large aromatic groups have small IUPAC representations, which helps with modeling highly conjugated systems. We’ll adjust the writing to avoid overstating the benefits of IUPAC names.

---

> ### Author Response · Authors · 2021-11-20
> **Response to R2 [2/2]**
>
> * “ ‘ignoring the embedding for “nitroso” ’ I would make this part of the sentence, rather than a footnote”
>
> We’ll move this to the main text. The original word2vec paper also ignored the query words when doing this analysis (arxiv:1301.3781, section 4 second paragraph).
> * “discarding the same from the statistics seems antithetic to the nature of a novelty metric”
>
> The novelty metric only really makes sense for de novo generation, not select-and-replace editing. We included it as a way to compare to prior work, but would be happy to remove it. If we didn’t remove cases where C5T5 regenerated the base molecule, our validity statistic would go up, and novelty would go down.
> * “How was synthetic accessibility computed? Is 0 high or low accessibility?”
>
> We got this backward -- thanks for catching this.
> * “Same for the logP increase, i see HierG2G producing a mean of 4 or so, and C5T5 is a little over 1?”
>
> HierG2G on average gets better logP increase than C5T5, which is unsurprising since C5T5 is carrying out the much more difficult task of replacing a specific part of the molecule to increase logP rather than being allowed to modify the input without constraints. However, given that we’re likely to give the user whichever molecules have the greatest logP increase, the max or e.g. 90%ile, where C5T5 outperforms HierG2G, is probably more important.
> * “there's essentially N squared pairs for the training library”
>
> Training pairs for HierG2G must be similar molecules. If HierG2G were trained on randomly selected pairs of molecules, then during inference time it would produce radically different molecules from the input. Figure 7 shows experimentally that the number of available training pairs decreases very rapidly as the required similarity increases.
> * “Are diphosphate and disulphate tokens?”
>
> There is no canonical way to tokenize IUPAC strings. We chose to tokenize by splitting names into base tokens using the SentencePiece algorithm, where the base tokens were taken to be all keywords from the IUPAC-processing Java library OpSin. Some of these “base tokens,” such as “diphosphate” and “disulphate,” could be further split, but because Opsin has these as recognized substrings, we use them as base tokens.

---

> > ### Comment · Reviewer_gkwv · 2021-11-29
> > **Dicussion period**
> >
> > I thank the authors for the thorough replies to all the reviewer's comments. I think the paper has improved somewhat with the additional discussion, but it is not yet at a level of theoretical novelty or applied performance gains to warrant acceptance, and issues such as comparing with more established baselines. I am revising my score up to a weak reject.

---

### Official Review · Reviewer_pKJo · 2021-11-02

**Correctness:** 4
**Technical Novelty And Significance:** 2
**Empirical Novelty And Significance:** 3
**Recommendation:** 5
**Confidence:** 3

**Main Review:**

**Comments**

Overall the paper presents an interesting treatment for the molecule property optimization (e.g., lead optimization, etc). The approach is simple but seems to be effective. The IUPAC naming alternative to SMILES can be of its own contribution to a broader context of chemical engineering.

However I have several concerns regarding the current draft.

1. The technique contribution is not very significant. The IUPAC naming is interesting but itself is not solid enough to be considered as technical innovation. The masked pretraining has been seen in other context like NLP, program language, etc. Nevertheless, I think it is ok for an application paper if the quality of experiments is high and the results are solid. So I would focus more on the experimental part.

2. The only baseline studied is Hierarchical G2G (Jin et.al) in Figure 6. Given that there has been many works in the space of molecule optimization, it would be necessary to include more baseline results, especially those with quantitative comparisons. Although some of the baseline methods may not be able to complete all the tasks, it would still be necessary to compare in the situations where baseline methods apply. In this way we can appreciate how significant the results are, especially given that the method proposed in this paper is self-supervised.

3. I would also like to learn more about the limitations of this method. For example, the IUPAC limits the possibility of generating new functional groups. There must be a trade-off between model capacity and the variance of results.

4. The extrapolation is not for free. I’d like to learn more about the failure cases where the model is not able to generate the molecules with desired properties.

5. (Optional) while the model presents an unsupervised way of learning, it would be interesting to see if it can be further fine-tuned with the paired molecule data for lead optimization. In this way, one can compare the results directly with G2G in their benchmarks.


**Summary Of The Paper:**

**Summary**

This paper proposed a way to modify the molecule based on language pretraining techniques. The sequence representation of molecules is based on IUPAC names, which can be more semantically meaningful and much easier to model than the SMILES or graph based molecule representation. The pretraining is done via a conditional text generation model where the model predicts the fragment names based on the remainder of the molecule and corresponding property values. The application on downstream molecule property optimization tasks show that the proposed approach is effective at obtaining high quality molecules.


**Summary Of The Review:**

**Review summary**

This is an interesting application paper in the domain of molecule optimization with some minor technical contributions. The preliminary results are interesting, but it would be more solid with more quantitative comparison with existing methods on existing benchmarks.

---

> ### Author Response · Authors · 2021-11-20
> **Response to R1**
>
> * “The masked pretraining has been seen in other context like NLP, program language, etc.”
>
> As detailed in the introduction, we believe our method has two significant sources of novelty. The first is the use of IUPAC names. Despite dozens of papers across machine learning & chemistry venues that investigate a variety of learning tasks on molecules, none have used IUPAC names as a base representation. Using IUPAC names lends a considerable amount of interpretability to any molecular learning task, and interpretability is especially important for lead optimization, where scientists are deeply involved in the optimization process. Even though the idea of using IUPAC names is not technically complex, we believe the fact that IUPAC names are both entirely unexplored in this community and have a clear interpretability advantage over other representations is a strong case for the novelty of our method.
>
> The second source of novelty is our proposed self-supervised objective. Many recent well-cited papers in NLP have mainly contributed novel masking strategies to improve self-supervised learning for text (e.g. SpanBert [doi:10.1162/tacl_a_00300], T5 [http://jmlr.org/papers/v21/20-074.html], whole word masking [arxiv:1906.08101]). However, our objective goes beyond merely proposing a new masking strategy: 1) C5T5 uses a novel conditional infilling objective that leverages masking in a new way to learn the joint distribution of property values and molecules. This form of masked pretraining has not been previously explored for sequence modeling, and we believe it is a considerable source of novelty in our submission that may be of interest not only for drug discovery but also for NLP, RL and other sequence modeling tasks. 2) Our conditional infilling objective leverages this learned joint distribution to enable zero-shot targeted edits. This means that our results are achieved entirely without access to pairs of edited molecules, which are challenging to acquire in large quantities and limited to known editing strategies.
>
> * “Given that there has been many works in the space of molecule optimization, it would be necessary to include more baseline results”
>
> See overall comment
> * “I would also like to learn more about the limitations of this method. For example, the IUPAC limits the possibility of generating new functional groups.”
>
> Any organic molecule can be represented with an IUPAC name, so there is no hard constraint on what C5T5 can output. In particular, new functional groups can always be represented using the IUPAC naming rules. However, as you point out, using IUPAC names makes it easier for the model to suggest edits that can be represented using a small number of IUPAC tokens. Similarly, the SMILES version of C5T5 has an easier time suggesting edits that can be represented with a small number of SMILES tokens -- i.e. that change only a small number of atoms. In this sense, the two representations have different strengths, so a system that suggests molecular edits to users should consider using both representations. Is there some specific discussion on the limitations of IUPAC that you think should be added to our existing discussion in Section 5?
> * “I’d like to learn more about the failure cases where the model is not able to generate the molecules with desired properties”
>
> C5T5 is designed to be used by humans who select specific parts of molecules to modify. Selecting, say, a comma between two consecutive locants would result in C5T5 being unable to come up with a better molecule. In practice, domain experts are likely to instead choose parts of the molecule that could plausibly be changed to improve the property value; it’s difficult to determine which failure cases result from an unoptimizable token being masked vs. C5T5 failing to find a good edit. What specifically would you like to understand better about failure cases?
> * “it would be interesting to see if it can be further fine-tuned with the paired molecule data”
>
> This is certainly possible, in much the same way that T5 is finetuned for text-to-text transfer tasks. However a key benefit of having a fully self-supervised approach is that it does not require pairs of similar molecules with different properties, which are either difficult to acquire (if using experimental data), or overly reliant on known editing strategies (if generating pairs from a matched molecular pair analysis).

---

> > ### Comment · Reviewer_pKJo · 2021-11-26
> > **RE: Response to R1**
> >
> > Thanks for your reply!
> >
> > Regarding the limitation of IUPAC, I think it would be good to mentioned that "Any organic molecule can be represented with an IUPAC name, so there is no hard constraint on what C5T5 can output." and the corresponding explanations in the paper.
> >
> > Regarding the failure cases, I'm more interested in the situation where the approach failed to extrapolate. For example, when <a molecule, a super good target property> is specified but it might be in general too hard to find such an edit. But this is more of a discussion rather than a concern, as it is understandable to have limited ability of extrapolation.
> >
> > As I mentioned that the technical contribution is not significant, thus I would focus more on the experimental contribution. I still feel it would be good to better position the paper among existing methods. For example, as the authors mentioned that it is possible to further fine-tune with the paired molecule data, it should be easy to perform comparison using the benchmark from HierG2G's paper.

---

### Author Response · Authors · 2021-11-20
**Overall Response**

Thanks for taking the time to review our submission! We respond to each reviewer individually below. In this overall comment, we hope to clarify the motivation behind C5T5 and how it differs from other work in generative models for drug design. The drug design process has many subtasks, and AI for drug design is still in its infancy, attempting to design approaches tailored to each of these subtasks. For example, many papers focus on generating molecules de novo (very early in drug development) or with a simple constraint that the molecule must include some scaffold (early on after a lead molecule has been identified). Benchmarks like GuacaMol and MOSES focus on these tasks. In contrast, we are concerned here with a crucial but later-stage part of the pipeline, where chemists seek to make targeted modifications to molecules to maintain their pharmacodynamic activity while improving their pharmacokinetics. This goal and its associated design constraints make it inappropriate to compare C5T5 to prior methods that evaluate on e.g. GuacaMol (as we explain in detail below). However, we believe this setting will be of interest to the ICLR community, and serve as a baseline in its own right to future work on interpretable targeted optimization.

Our method does not address the same problem domain as GuacaMol or MOSES. In fact, there is no way to run C5T5 on any of the GuacaMol or MOSES benchmarks. R2 mentions the scaffold-based tasks in GuacaMol. There are two scaffold-based goal-directed benchmarks in GuacaMol: “scaffold hop” and “deco hop.” Neither of these benchmarks considers optimizing chemical property values, which is the task that C5T5 addresses. In addition, the scaffolds in these datasets are represented as SMARTS strings, which do not always correspond exactly to specific IUPAC tokens. The GuacaMol benchmarks that do consider optimizing chemical property values are de novo benchmarks. C5T5 is incapable of generating molecules de novo, in much the same way that the de novo methods that evaluate on GuacaMol and MOSES are incapable of select-and-replace editing.

Comparing to genetic algorithms or to other methods that do constrained optimization at inference time is more feasible. There are a few reasons we don’t make these comparisons.

* Most such baselines are in fact performing a different task. Specifically, methods that do constrained optimization on molecular graphs constrain the output to contain a certain subgraph. This allows the method to append to the graph anywhere. In contrast, C5T5 is limited to edit only the part of the molecule specified by the IUPAC mask. This is more desirable for any sort of expert guided editing, as is commonly done in lead optimization.
* In the same vein, most IUPAC masks cannot be translated cleanly to SMILES or graph masks, even if the IUPAC mask contains only functional groups tokens. For example, the “acetyloxy” in “2-acetyloxybenzoic acid” does refer to a specific set of atoms, but because the “2-” is unmasked, masking “acetyloxy” does not correspond to masking those atoms in a SMILES or graph representation.
* Optimization methods often aim to minimize or maximize a certain score, but C5T5 only seeks to get property values within some range. For example, R2 wonders whether C5T5 would find pentatritylbenzene, presumably given e.g. ditritylbenzene with “di” masked. We would expect a logP-optimizing method to find pentatritylbenze, but C5T5 does not (when given that input & mask), perhaps because pentatritylbenzene is so far out of the PubChem distribution, and the loss makes no distinction between logP values slightly above the cutoff and very far above the cutoff.
* Methods that optimize the property predictor at inference time have a potentially undesirable advantage over methods that do not. As R2 mentions, requiring a property predictor at inference time poses no significant computational cost in the case of properties like solubility, but it does pose other problems. First is that it’s difficult to determine if the method is overfitting to the property predictor. Using one property predictor for optimization and another for evaluation may help, but adversarial example research has shown that examples that trick one model often trick other unseen models too (arxiv:1312.6199). A second disadvantage is that inference-time optimization methods can’t be used to optimize expensive-to-compute properties, and they can’t be trained on experimental data. We have not yet trained C5T5 on either expensive properties or experimental data, but we believe for this reason that it is worth pursuing research on methods like C5T5 that don’t optimize a property predictor.

---

### Decision · Program_Chairs · 2022-01-20

**Decision:**

Reject

**Comment:**

This work proposes to use a transformer model and language model inspired self-supervised training techniques to generate local modifications of organic molecules. The use of IUPAC names coupled with language inspired pre-training is indeed an interesting idea worthy of exploration. The paper has a lot of promises in this regard but needs more work to deliver it through the finish line. In the rebuttal, the authors have provided strong arguments toward the advantages of using IUPAC representation. While these arguments make sense, they are more or less conceptual and better and more clear empirical evidences are required to back them up.